# Genome-Wide Selection Sweep between Wild and Local Pigs from Europe for the Investigation of the Hereditary Characteristics of Domestication in Sus Scrofa

**DOI:** 10.3390/ani12081037

**Published:** 2022-04-15

**Authors:** Yiming Gong, Hao-Yuan Zhang, Ying Yuan, Yongmeng He, Weiyi Zhang, Yanguo Han, Risu Na, Yan Zeng, Jia Luo, Haili Yang, Yongfu Huang, Yongju Zhao, Zhongquan Zhao, Guang-Xin E

**Affiliations:** College of Animal Science and Technology, Southwest University, Chongqing 400716, China; g18080010835@163.com (Y.G.); swuzhanghy@163.com (H.-Y.Z.); 15703001957@163.com (Y.Y.); yongmenghe123@163.com (Y.H.); lwing0829@163.com (W.Z.); hyg2015@swu.deu.cn (Y.H.); narisu@swu.edu.cn (R.N.); zengyan@swu.edu.cn (Y.Z.); b20161705@swu.edu.cn (J.L.); yhlhouni@163.com (H.Y.); h67738337@swu.edu.cn (Y.H.); zyongju@163.com (Y.Z.); zhongquanzhao@126.com (Z.Z.)

**Keywords:** European, domestic pigs, wild boar, GWSA, genetic contribution

## Abstract

**Simple Summary:**

Wild boar (WB) originate in southeastern Asia. With the migration of WBs into the European continent, their domestication started as early as the Neolithic period. With the continuous domestication and considering people’s different needs for pork, many local WB breeds with their own characteristics appeared on European land. In this article, the whole genome files of 10 local WB and 38 domestic local breeds (1098 individuals in total) were screened in terms of genetic differences in metabolism, disease, and temperament by genome-wide association scan selection signals. Moreover, the geographical distribution and historical events of the different breeds were combined to confirm the wide genetic contribution of WB to the domestication process. The results of this study will provide a reference on the metabolism and emotionality between European WBs and domestic pigs.

**Abstract:**

The phenotypic characteristics of existing domestic pigs (DPs) greatly differ from those of wild boar (WB) populations thousands of years ago. After thousands of years of human domestication, WB and DP have diverged greatly in terms of genetics. Theoretically, worldwide local pigs have independent contributions from their local WBs at the beginning of Sus scrofa domestication. The investigation of the vicissitude of the heredity material between domestic populations and their wild ancestors will help in further understanding the domestication history of domestic animals. In the present study, we performed a genome-wide association scan (GWSA) and phylogeny estimation with a total of 1098 public European Illumina 60K single nucleotide polymorphism data, which included 650 local DPs and 448 WBs. The results revealed that the phylogenetic relationship of WBs corresponds to their geographical distribution and carries large divergence with DPs, and all WB breeds (e.g., *HRWB*, *SBWB*, and *TIWB*) presents a closely linkage with the middle WB (e.g., *HRWB*, and *PLWB*). In addition, 64 selected candidate genes (e.g., *IDH2*, *PIP5K1B*, *SMARCA2*, *KIF5C*, and *TJP2*) were identified from GWSA. A total of 63 known multiple biological functional pathways were annotated by 22 genes, and ubiquinone and other terpenoid-quinone biosynthesis pathways that belong to the metabolism of cofactors and vitamins were significantly enriched (*p* < 0.05). The most frequent (28.57%) pathways were classified under metabolism. We confirmed that the middle European WB has made an important genetic contribution to the entire European WB populations. A series of selected genes discovered from this study provides the scientific community with a deeper understanding of the heredity performance of metabolism and emotion and the real purpose behind domestication.

## 1. Introduction

As important economic animals in meat production for human consumption in the world, domestic pigs (DPs) are important for achieving political and economic stability [1,2]. The DP population that is distributed widely throughout the world has excellent environmental adaptability and disease resistance [3,4], and it is the only genetic material library for commercial pig breeding [5]. Crossbreeding between WB and DP is becoming a favorable method for the improvement of some DP traits.

Many studies have identified the key candidate parameters for environmental adaptability and economic traits, such as high-altitude adaptability [6], litter size [7,8], growth performance [9,10], and meat quality [7]. In addition, the genetic diversity and population phylogeny classification of DPs are among the concerns of the academic community [4,11].

WB first appeared in Southeast Asia and then gradually spread to Asia and Europe. Since the Neolithic Age, humans have begun to domesticate WB populations [12,13], and around 6000 BC, European WBs were domesticated from the southeast of Europe and gradually expanded to northern European lands [14]. Around 4000 BC, with the expansion of the wild boar (WB) population from the Near East, data show that WBs were domesticated in the Paris Basin [15]. In some areas of northern Germany, data show that they had access to DPs [16].

WBs are inferior in terms of fecundity [17], growth ability, and meat quality [15] to DPs [17,18], although these characteristics are more advantageous in terms of disease resistance and adversity adaptability [19,20,21]. However, more research has focused on observing the dynamic distribution of WB populations and conservation [22,23,24].

At present, WB, as a global distributed ancestor of DP, can be regarded as homologous ancestors to help us understand the genetic material changes of DP from WB to domestic population in the past thousands of years. Therefore, in the present study, we conducted a wide-genome selected sweep analysis between WB and DP from Europe by using the public genome-wide single nucleotide polymorphism (SNP) data. In-depth investigation of the common genetic basis behind the common domestication activities of pig should be conducted to support and improve the understanding of the historical role of domestication.

## 2. Materials and Methods

### 2.1. Ethics Statement and Sample Collection

A total of 1098 European pigs Illumina 60k SNP data (Appendix A), including 448 WBs and 650 DPs, were obtained [25,26]. The regional distribution of 1098 individual European pigs is shown in Figure 1. The original datasets were amalgamated and filtered by Plink with minor allele frequency ≤0.05, and 41,748 SNPs were retained for subsequent analysis.

### 2.2. Data Analysis

A neighbor-joining phylogenetic tree was estimated using VCF2Dis (https://github.com/BGI-shenzhen/VCF2Dis, accessed on 7 July 2021), visualized using FastME 2.0 [27], and subjected to beautification by using iTOL (https://itol.embl.de/, accessed on 8 July 2021). Principal component analysis (PCA) was performed and visualized by GCTA and R program (ggplot2 package), respectively. The pairwise fixation index (*F_ST_*) [28] and Tajima’s D [29] were calculated with 40 kb sliding windows by using vcftools (http://vcftools.sourceforge.net/, accessed on 25 June 2021) between DPs and WB. Candidate genes were annotated by the intersection of both parameters with top 10% threshold (*F_ST_* > 0.2052, Tajima’s D > 4.4681). The results are shown in Appendix A. Finally, genome annotation based on Kyoto Encyclopedia of Genes and Genomes was carried out for the candidate genes by using KOBAS 3.0. The linkage disequilibrium of single candidate genes is displayed using HaploView 4.2 (https://www.broadinstitute.org/haploview/haploview, accessed on 26 June 2021).

## 3. Results and Discussion

According to the classification of the habitat large geographical plate (11 redefined regions) of breeds (Figure 2A, Appendix A), the phylogenetic relationship of all European individuals revealed large-scale exchange of genetic material. However, a large genetic divergence exists between the DP and WB populations in Europe.

All WBs formed a tight cluster, including several DPs from southern, middle, and western Europe. Several WBs from middle Europe have a close relationship with almost all groups of wild populations geographically located in Europe. Consistent results were reproduced and verified by PCA (Figure 2B) by using the spatial distribution of European WB (Figure 2C). Thus, middle European WB have important heredity contributions to the phylogeny of the entire European WBs. Almost all the maternal lines in northern Europe originated from varieties obtained by domestication in the south of middle Europe [30] This assumption supports that central Europe may have been a center for early European pig domestication [31].

Human activities have greatly restricted the habitat boundary of WBs, leading to the well-defined genetic structure classification of WBs [32]. The local gene structure of European WBs may have been artificially altered by severe (or non-random) exploitation for hunting purposes and captive breeding of animals, among other behaviors [33]. However, the number of European WBs has been increasing rapidly because of the lack of large natural enemies and reduced hunting. This condition has caused a conflict between WB and human society as a result of the intensified resource occupation of large-size population and eventual forced migration of WBs [34,35]. As early as the 16th century, considering the expansion of WBs, crossbreeding between wild and DPs often occurred [36]. This phenomenon not only explains the close relation between a certain proportion of middle European DPs and WBs, but also implies that the high genetic diversity within middle European WB might have been caused by the extensive exchange of heredity material in WB migration. Studies support that middle European WBs have a great genetic contribution to European WBs [32,37,38,39].

Based on the results of the GWSA (Figure 3), 43 windows were obtained from the interaction of the top 10% windows of both parameters (*F_ST_* > 0.2052, Tajima’s D > 4.4681). Furthermore, 64 overlapping candidate genes (e.g., *ALDH1A3*, *CSNK2A2*, *SMARCA2*, and *PARP1*) were identified from the 43 interacted windows. In addition, the results show that 22 of 64 genes were enriched in 63 known multiple biological functional signal pathways (Appendix A), such as inositol phosphate metabolism, nuclear factor-kappa B signaling pathway, vascular smooth muscle contraction, and the Wnt signaling pathway. High frequencies of pathways were classified into metabolism (28.57%), diseases (19.04%), and environmental information processing (15.87%).

Six genes, namely *ALDH1A3*, *COQ6*, *ENTPD5*, *IDH2*, *PISD*, and *PIP5K1B*, were enriched in a series of metabolism-related pathways. These selected genes are related to amino acid metabolism and degradation of foreign substances. *ALDH1A3* and *ENTPD5* can promote glucose metabolism [40] and regulation of phosphate levels in the body, respectively [41]. In addition, *IDH2* and TCA (tricarboxylic acid cycle) are closely related, while *IDH2* inhibitors (AGI-6780) can substantially reduce the activity of the TCA and ATP levels [42]. The dimer formed by *IDH2* and *IDH1* can catalyze the reversible NADP+-dependent oxidative decarboxylation of isocitrate to α-KG [43]. The addition of α-KG to pig diets can remarkably reduce the apparent digestibility of calcium and phosphorus [44]. Thus, DPs have evolved to have more powerful feeding and digestion capabilities with different feeding methods and food types compared with WBs in the long-term domestication process.

Seven genes (e.g., *ALDH1A3*, *CSNK2A2*, *DNAH11*, *SMARCA2*, *SS18*, *TNR*, and *PIP5K1B*) have been identified as immune-related in processes such as the formation of cancer, bacterial infections, and viral infections. *ALDH1A3* plays an important role in the metastasis of pancreatic cancer [40] and the pathogenesis of prostate cancer [45]. In particular, *ALDHA3* can be used as a metabolic target for cancer diagnosis and treatment [46]. *CSNK2A2* is associated with systemic lupus erythematosus (SLE) in previous studies [47]. SLE, as a representative disease of over-immunity, shows the important role of *CSNK2A2* in the immune system. In addition, WBs are widely considered to have good disease resistance and immunity because of the long-term symbiosis with pathogens, such as African swine fever [48] and influenza A [49]. Thus, the results of this study are important for the further understanding of the genetic basis of the differences in immunity and disease resistance between WBs and DPs.

*SMARCA2* plays an important role in porcine cleavage cells [50]. Notably, *SMARCA2* and *SMARCA4* alternately occupy the catalytic sites of the SWI/SNF chromatin remodeling, completely change the transcription of *Sm2*, and affect embryonic development [50]. The litter size of WBs is generally lower than that of DPs. For example, only 6.3 corpus luteum per ovulation period were observed in Polish WBs, and the average litter size is 5.9 [51]. By contrast, during the 1980s in France, the average litter size of domestic sows was between 15 and 25 per year [52]. Therefore, *SMARCA2* is related to litter size and is domesticated from WBs to DP.

Obviously, DPs and WBs have great differences in terms of temperament. The genome of the DP is strongly selected for loci affecting behavioral and morphological correlates. [53] Damage to piglets by sows is a common barbaric behavior [54,55,56,57]. Savagery has also been observed in farmed WB sows, and Baxter et al. found that in strains selected for high piglet survival (as a selective trait), WB breeds from outdoor breeding are highly aggressive when bred indoors [58]. In DPs, they maintain dominance and control over food through various repeated fighting behaviors [59]. The heritability of aggressive behavior at mixing ranges from 0 to 0.44 in weaners, growers, replacement gilts, and mature sows [60,61,62,63,64]. In the study, multiple pathways and related genes related to emotions, such as *KIF5C*, are mainly present in neuronal cells [65]. *KIF5C* can transport signals to the nerve terminal in a speed of ~1 µm/s [66]. Especially, the phosphorylation of *KIF5* as the precursor of *KIF5C* generally increases, and this phenomenon is associated with the organelle transport increase in axons [67]. In the present study, we discovered that the SNPs from the *KIF5C* coding region of DPs showed a tighter linkage than that of WB populations (Figure 3B). In addition, *COQ6* is associated with nephrotic syndrome, sensorineural hearing loss, and neurotrophy [68]. WBs are highly vigilant against hunting from natural enemy and humans, whereas DPs benefit from human activities and do not exhibit a strong antagonism. Coincidentally, changes have been observed in animal temperament caused by selection pressure during domestication [69]. Therefore, *KIF5* and *COQ6* genes may be related to the high vigilance and social sensitivity of WBs.

## 4. Conclusions

Our research results showed that the domestication of pig is a hidden heredity black box, rather than representing the simple requirement from human social development. The large number of genes related to adaptability, metabolism, and emotional sensitivity discovered in this study will help us understand the most important genetic basis in animal domestication.

## Figures and Tables

**Figure 1 animals-12-01037-f001:**
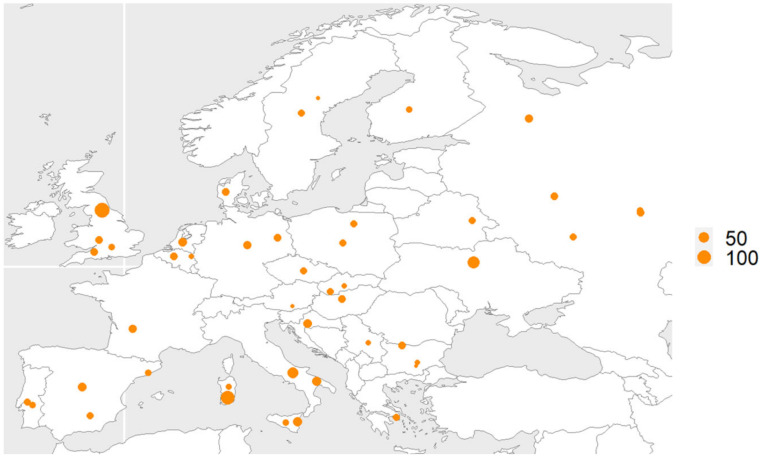
Regional distribution map of 1098 individual European pigs. Refer to the data source article for specific group geographic divisions [22,23]. These dots indicate the distribution, source and quantity of samples.

**Figure 2 animals-12-01037-f002:**
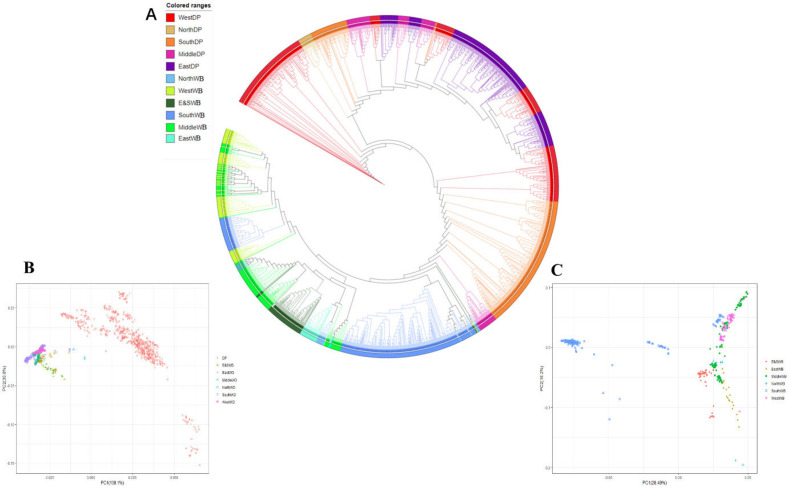
Population genetic network and PCA of European WBs and DPs. (**A**) Classification of the habitat large geographical plate of European WBs and DPs. Each color represents the domestic and WB population in a different area, including WestDP, NorthDP, SouthDP, MiddleDP, EastDP, NorthWB, WestWB, E&SWB, SouthWB, MiddleWB, EastWB, DP and WB, which was divided by regions; (**B**) PCA of the 60K SNP data set: based on all the available data (1098), divided into seven species by region; (**C**) Principal component analysis of the 60K SNPs data set: based on the European WB data divided by six regions.

**Figure 3 animals-12-01037-f003:**
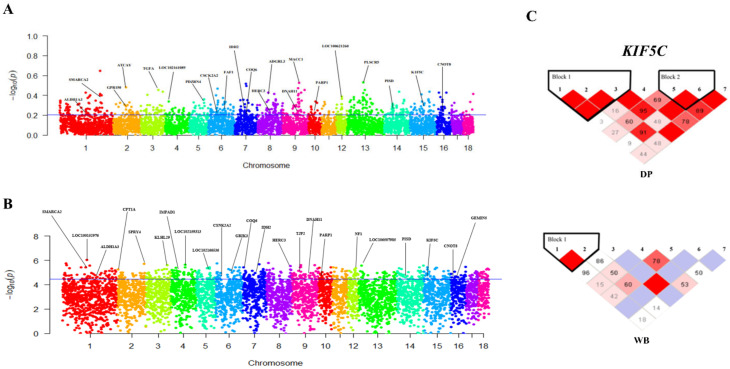
Genome-wide selection scan for SNPs in European WB and DP by using pairwise fixation index (*F_ST_*) and Tajima’s D. (**A**) Manhattan map of *F_ST_* between groups; (**B**) Manhattan map of Tajima’s D between groups; (**C**) Result of genetic linkage of *KIF5C* by linkage disequilibrium.

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
