# Peer review of "Genome-Wide Selection Sweep between Wild and Local Pigs from Europe for the Investigation of the Hereditary Characteristics of Domestication in Sus Scrofa"

_animals, 2022, doi:10.3390/ani12081037_

Round 1
Reviewer 1 Report
Summary
The authors searched for genetic regions under selection in domestic pigs (DPs) and wild boars (WBs). For this purpose, there were available 650 European domestic pigs and 448 European wild boars with 60k SNP data. They performed phylogenetic analysis and computed FST and Tajima’s D for selection detection. They find 64 candidate genes and interpret corresponding pathways and function.
General comments
Although domestication of pigs is an interesting subject, this manuscript shows severe methodological concerns.
Comments
- In general, the study is not described in sufficient detail. Much information is missing which is mandatory to understand the methodology and results.
- 2 Data analysis
In its present form, the information in this section is too vague to understand what was done.- How was FST and Tajima’s D calculated exactly? Pairwise between all animals, within DPs and WBs or between DPs and WPs?
- The software VCF2Dis is cited. I could not find a publication for this software and from the very short github page the methodology is not clear. Only published software should be used.
- Describe the method and parameters of the neighbour-joining phylogenetic tree estimation.
- What was the basis for the principal component analysis (PCA)? Are the underlying SNPs LD-pruned (they should be)? How many principal components were used for the analysis? In Fig. 2 B and C there are percentages added to PC1 and PC2. What do these mean and how can they be above 100%?
- The criterion for selection windows is “the interaction of top 10% windows of both parameters (FST>0.2052, Tajimas’D>4.4681)”. There is no statistical justification presented for this threshold. What is the statistical null hypothesis? Even if no selection is present, there will always be windows in the top 10% range.
- The geographic regions used in the manuscript are not defined.
- 1 Ethics Statement and Sample Collection
- Which Illumina SNP kit exactly was used?
- Only very basic quality control is reported. What about callrate, Hardy-Weinberg equilibrium, relatedness and homozygosity?
- FST and Tajima’s D are very basic methods for detecting selection. These should be followed up with more up-to-date methods. Otherwise the results are not convincing.
- Present the complete results (possibly in the supplement), especially the windows and genes of interest found in this study with corresponding FST and D values.
- 3 A and B: On the y-axis p values are shown. How were these computed? Was adjustment for multiple testing performed are integrated in the statistical test?
- The authors present results of an enrichment analysis. Add detail how this analysis was performed and how multiple testing was addressed. In the table in the supplementary analysis all p values are much larger than 0.05.
- Important references are not mentioned, e.g.
- Larson, G. et al. Ancient DNA, pig domestication, and the spread of the Neolithic into Europe. Proc Natl Acad Sci USA 104, 15276–15281 (2007).
- Krause-Kyora, B. et al. Use of domesticated pigs by Mesolithic hunter-gatherers in northwestern Europe. Nat Commun 4, 2348 (2013).
- Frantz, L. A. et al. Evidence of long-term gene flow and selection during domestication from analyses of Eurasian wild and domestic pig genomes. Nature genetics 47, 1141–1148 (2015).
- Price, M., Hongo, H. The Archaeology of Pig Domestication in Eurasia. J Archaeol Res 28, 557–615 (2020). https://doi.org/10.1007/s10814-019-09142-9
- Caliebe A, Nebel A, Makarewicz C, Krawczak M, Krause-Kyora B. Insights into early pig domestication provided by ancient DNA analysis. Sci Rep. 2017;7:44550. Published 2017 Mar 16. doi:10.1038/srep44550
- Ai, H., Fang, X., Yang, B. et al. Adaptation and possible ancient interspecies introgression in pigs identified by whole-genome sequencing. Nat Genet 47, 217–225 (2015). https://doi.org/10.1038/ng.3199
- The English language should be improved. Some sentences are not understandable or include grammar error. Examples
- “Wide-genome” should be “genome-wide” (even wrong in the title)
- “As early as the 16th century, the WB formal mating rights frequently appeared in the DP populations [29].” Not clear to me.
- “Damage to piglets by sows is a common barbaric behavior, with up to 15% of newborn sows barbaric to their offspring [47-50].” Same.
Author Response
Dear Reviewer:
Thanks for your suggestion! We have revised our manuscript and answer your questions. Please check the Word below.

Reviewer 2 Report
Please find my suggestions below:
I have used parenthesis for citing the text of the manuscript.
’…reference on metabolism and emotionality between European wild boars and domestic pigs for future researches…’
'emotionality'? Is not behaviour a better word for that?
’wide-genome selected sweep analysis (WGSA)’
That’s weird, causes confusion. Additionally, authors are using GWSA (see Results section) as well… It is too similar to whole-genome association scan (WGAS) or genome-wide association scan (GWAS). Something else should be used instead of WGSA. Authors also use ’ Genome-wide selection scan…’ in the MS. Please make an order regarding the terms.
’… We have verified once again that the …’ What does that ’once again’ mean?
’…adversity adaptability…’ Please rephrase
I would not call wildboar a ’natural fossil’. It is not.
’…of the mysterious common genetic basis…’ Please omit ’mysterious’.
’…were maintained for subsequent analysis…’ I suggest: …were retained…
’…classification of the habitat large geographical plate (11 redefined regions) of breeds (Figure. 2A)…’ It is not understandable. At least not for me.
’… were identified from the 43 interacted windows…’ Please explain!
In Materials and Methods, groups should be defined. I just presume, that DP and WB were the two groups. It seems, there are hints in the texts, that other grouppings were also used.
’… Therefore, it is not difficult to understand that…’ This part can be deleted.
’… we firmly believe that…’ Please delete!
At the end, there are just two question to the authors:
Have they found selection sweep around ADCY1 gene? Articles from 2021, 2022, are suggesting association with muscle development and meat quality in DP or association with behaivour in WB.
Have the authors tried a third or more sweep finding tools to refine intersections of the top hits?
After revision, I suggest the MS for publication.
Author Response

(The authors gave the same response as above.)
